# MuSeD: A Multimodal Spanish Dataset for Sexism Detection in Social Media Videos

**Laura De Grazia**[1], **Pol Pastells**[1], **Mauro Vázquez Chas**[1], **Desmond Elliott**[2],
**Danae Sánchez Villegas**[2], **Mireia Farrús**[1], **Mariona Taulé**[1]
[1]University of Barcelona, CLiC-Language and Computing Center
[2]University of Copenhagen, Department of Computer Science
{lauradegrazia, pol.pastells, mauro.vazquez, mfarrus, mtaule}@ub.edu
{de, davi}@di.ku.dk

## Abstract

Sexism is generally defined as prejudice and discrimination based on sex or gender, affecting every sector of society, from social institutions to relationships and individual behavior. Social media platforms amplify the impact of sexism by conveying discriminatory content not only through text but also across multiple modalities, highlighting the critical need for a multimodal approach to the analysis of sexism online. With the rise of social media platforms where users share short videos, sexism is increasingly spreading through video content. Automatically detecting sexism in videos is a challenging task, as it requires analyzing the combination of verbal, audio, and visual elements to identify sexist content. In this study, (1) we introduce MuSeD, a new Multimodal Spanish dataset for Sexism Detection consisting of $\approx 11$ hours of videos extracted from TikTok and BitChute; (2) we propose an innovative annotation framework for analyzing the contributions of textual, vocal, and visual modalities to the classification of content as either sexist or non-sexist; and (3) we evaluate a range of large language models (LLMs) and multimodal LLMs on the task of sexism detection. We find that visual information plays a key role in labeling sexist content for both humans and models. Models effectively detect explicit sexism; however, they struggle with implicit cases, such as stereotypes—instances where annotators also show low agreement. This highlights the inherent difficulty of the task, as identifying implicit sexism depends on the social and cultural context.[1] **Warning**: *This paper contains examples of language and images which may be offensive.*

## 1 Introduction

Sexism is a complex phenomenon, generally defined as prejudice or discrimination based on sex or gender.[2] The effects of sexism are systemic (Javidan, 2021), spreading across every sector of society, from the macro-level (social institutions) to the meso-level (social interactions) and the micro-level (internalization of sexist beliefs). Studies on sexism conceptualize it as an ideology (O'Brien, 2009) that reproduces itself by portraying women as naturally inferior, and by promoting heterosexuality and cisgender identity as normative models. In doing so, the ideology of sexism marginalizes subjectivities that do not conform to these norms, such as intersex and transgender individuals, by portraying them as abnormal. Social networks provide a crucial setting for analyzing the forms and narratives through which sexism spreads and, in doing so, for developing strategies to counter it (Are, 2020; Simpson & Semaan, 2021). Sexism can be conveyed on social media platforms in multiple ways, not only through text but also through a combination of modalities such as text and

---

[1]The dataset annotations will be made available for research purposes at https://github.com/lauradegrazia/MuSeD_sexism_detection_videos

[2]https://www.britannica.com/topic/sexism

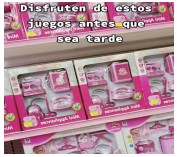 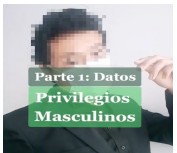 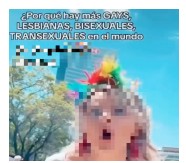 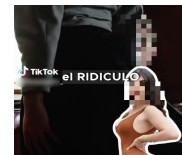

|  (a) Stereotype | (b) Inequality | (c) Discrimination | (d) Objectification |

| Sexist case | Original text | English translation |
|---|---|---|
| Stereotype (a) | Mi gente, mi gente, miren, **disfruten de estos juegos que traen plancha, lavadora, que traen así, porque las feministas pronto van a protestar que las niñas no deben jugar con esto**. | My people, my people, look, enjoy these toys, **enjoy these games that come with an iron, a washing machine, that come like this, because the feminists will soon protest that girls shouldn't play with this**. |
| Inequality (b) | Es hora de admitirlo, **los hombres tenemos privilegios por sobre a las mujeres, sociales y legales** [. . . ]. **Tenemos seis años más de pena por el mismo delito cometido por una mujer y el doble de posibilidades de ser encarcelado.** | It's time to admit it, **men have privileges over women, both socially and legally** [. . . ]. **We have six more years of punishment for the same crime committed by a woman and twice the chances of being incarcerated.** |
| Discrimination (c) | **¿Por qué hay más gays, lesbianas, bisexuales, transexuales en el mundo? Porque les vendieron la mentira de la identidad sexual** para pertenecer a un grupo que ofrece la liberación sexual como solución a los problemas existenciales [. . . ]. | **Why are there more gays, lesbians, bisexuals, and transsexual in the world? Because they were sold the lie of sexual identity** to belong to a group that offers sexual liberation as a solution to existential problems [. . . ]. |
| Objectification (d) | **¿Cómo ser gracioso y conquistar sin hacer el ridículo?** Mi hermano, una habilidad clave para ser gracioso y atractivo es la malinterpretación intencional. | **How to be funny and win women over without making a fool of yourself?** My brother, a key skill to being funny and attractive is intentional misinterpretation. |

Table 1: Examples of sexist videos along with their text transcriptions. Example (a) contains a stereotype, reinforcing traditional gender roles for women. Example (b) critiques the idea of *male privilege*, arguing that men are marginalized. Example (c) discriminates based on sexual orientation. Example (d) objectifies women by depicting strategies to "conquer" them.

image (memes) or text, audio and image (videos). This highlights the critical importance of analyzing sexism from a multimodal perspective.

The automatic detection of sexism on social platforms is a crucial task because it supports the work of human moderators, who have been shown to develop PTSD and depression (Newton, 2020), and helps prevent the spread and normalization of toxic narratives. It is also a challenging task, particularly when sexism appears in multimodal forms, where multiple modalities contribute to the production of discriminatory content. The identification of sexist content has primarily focused on text, particularly extracting data from X (Rodríguez-Sánchez et al., 2020; Samory et al., 2021; Plaza et al., 2023). Investigating sexism across multiple modalities is an increasingly relevant topic due to the growing proliferation of multimedia content online. The majority of studies in the area of multimodal sexism have focused on analyzing memes—images that combine pictorial content with overlaid text—and examining hate speech in videos (Chhabra & Vishwakarma, 2023). While previous work on identifying sexism in videos has primarily focused on sex-based sexism, particularly targeting women (Arcos & Rosso, 2024), the analysis of sexism that also encompasses sexual orientation and gender identity (O'Brien, 2009) remains largely unexplored.

In this work, we aim to address the following research questions: **Q1:** How can we create a systematic annotation framework to analyze the impact of textual, vocal, and visual modalities in conveying sexist or non-sexist content? **Q2:** To what extent does visual information play a key role in identifying sexism for both humans and models? **Q3:** How do model misclassifications relate to human disagreement? Our work makes the following contributions: **1**. We introduce MuSeD, a Multimodal Spanish Dataset for Sexism Detection, expanding the definition of sexism to encompass discrimination based on sex, sexual orientation, and gender identity. MuSeD includes content that discriminates based on sex, such as portraying women through stereotypes; sexual orientation, by including content that discriminates against non-heterosexual identities; and gender identity, by incorporating content that discriminates against transgender individuals. Table 1 presents representative

| | Platform | Language | Target group | Annotation of modalities | Annotator expertise |
|---|---|---|---|---|---|
| **Memes** | | | | | |
| Fersini et al. (2019) | FB, TW, IG, R | eng | W | ✗ | Crowd Sourcing |
| Fersini et al. (2022) | TW, R, MGTOW | eng | W | ✗ | Crowd Sourcing |
| Singh et al. (2024) | FB, IG, R, P | hin, eng | W | ✗ | Trained annotators |
| Ponnusamy et al. (2024) | IG, FB, P | tam, mal | W | ✗ | Trained annotators |
| Plaza et al. (2024) | Google image | eng, spa | W | ✗ | Crowd Sourcing |
| **Videos** | | | | | |
| Arcos & Rosso (2024) | TT | eng, spa | W | ✗ | Trained annotators |
| MuSeD (Ours) | TT, BitChute | spa | W, LGBTQ+ | ✓ | Moderator, one expert annotator, trained annotators |

Table 2: A comparison of existing multimodal datasets for annotating sexism and misogyny online. MuSeD comprises Spanish-language videos from TikTok and BitChute and introduces modality-specific annotations for sexism detection. W: Women, FB: Facebook, TW: Twitter, IG: Instagram, R: Reddit, P: Pinterest, TT: TikTok.

examples in our dataset. **2**. We propose an innovative annotation framework for labeling *sexist* and *non-sexist* content, with annotations conducted at multiple levels—(i) text, (ii) audio, and (iii) video (text, speech, and image)—to examine the contribution of visual and multimodal information to the identification of content as *sexist* or *non-sexist*. This is the first work to examine sexism in multimodal data using such a multi-level annotation scheme. **3**. We evaluate LLMs and multimodal LLMs on sexism detection using text-only and multimodal inputs, analyzing their alignment with human annotations. We find that visual cues help in labeling sexism, yet models struggle with implicit cases like stereotypes, where annotator agreement is lower. This highlights the challenge of detecting sexism in multimodal content.

## 2 Background and related work

### 2.1 Data resources for multimodal sexism

**Image resources** Currently, memes are the most commonly used source of multimodal datasets for detecting sexism and misogyny against women. The first study that analyzed sexism in memes was conducted by Fersini et al. (2019), who released the MEME dataset, consisting of 800 memes with *sexist* and *non-sexist* content. The dataset includes four subcategories: *Shaming*, *Stereotype*, *Objectification*, and *Violence*. Misogyny, closely related to sexism (Fontanella et al., 2024), is a subtype of hate speech that involves *hateful content against women* (Zeinert et al., 2021). A significant contribution to the detection of misogyny in memes came from *SemEval-2022 Task 5: Multimedia Automatic Misogyny Identification (MAMI)* (Fersini et al., 2022). The MAMI benchmark dataset includes 10,000 memes for training and 1,000 memes for testing, categorized into *Shaming*, *Stereotype*, *Objectification*, and *Violence*. The MIMIC dataset (Singh et al., 2024) and the MDMD dataset (Ponnusamy et al., 2024) contribute to research on misogyny detection in memes featuring low-resource languages. The EXIST 2024 dataset includes *sexist* and *non-sexist* memes in both English and Spanish (Plaza et al., 2024). It contains 2,000 memes per language for the training set and 500 memes per language for the test set.

**Video resources** The development of datasets for detecting sexism in videos is a new and promising field of investigation. Current research is mostly focused on the detection of hate speech, a definition that includes the use of discriminative and hateful content against individuals or groups based on a broad range of factors, including sexual orientation and gender identity (Chhabra & Vishwakarma, 2023). Das et al. (2023) used BitChute, a platform

primarily used to spread far-right conspiracies and hate speech (Trujillo et al., 2020), for collecting videos. They released HateMM, a multimodal annotated dataset of 1,083 videos ($\approx$ 43 hours), 431 labeled as hate and 652 as non-hate. Finally, Arcos & Rosso (2024) used TikTok to create a dataset in English ($\approx$ 12 hours) and Spanish ($\approx$ 14 hours), following the annotation framework developed by Plaza et al. (2023).

**Limitations of existing datasets**    Table 2 compares existing datasets for analyzing sexist and misogynistic content online. We observe that the primary focus of previous work is the analysis of sexism in memes, considering the relationship between text and image. The key differences between our study and previous work are: (i) we broaden the definition of sexism to include discrimination against a target group based on sex, sexual orientation, or gender identity (O'Brien, 2009), whereas previous work primarily focused on sexism based solely on sex; (ii) we select two different platforms as data sources: TikTok, a moderated platform, and BitChute, a low-moderation platform, which enables us to find diverse material; (iii) the annotation task is conducted across different modalities to evaluate the impact of each modality on identifying sexist content. Furthermore, the task involves selecting segments in which annotators identify sexism in videos. This method allows us to precisely identify where sexism occurs, providing a more accurate annotation; (iv) our annotation team was moderated by an expert on sexism in social media and included an experienced annotator specializing in the annotation of discriminatory content.

## 2.2    Multimodal models for sexism detection

Detecting social biases, including sexism, has traditionally relied on text-based models (Lei et al., 2024), which perform well in detecting explicit biases but struggle with implicit cues, sarcasm, and context-dependent expressions (Ao et al., 2022; Tiwari et al., 2023; Sánchez Villegas et al., 2024; Hee et al., 2024). In contrast, multimodal approaches have improved performance in sentiment analysis, sarcasm detection and hate speech detection, especially in cases where gestures, tone, and images shape interpretation (Bagher Zadeh et al., 2018; Sánchez Villegas, 2023; Tang et al., 2024; Arya et al., 2024). This is particularly relevant in sexism detection, where visual cues can reinforce harmful stereotypes beyond what is stated in text (Rizzi et al., 2023). A major challenge in multimodal learning is the reliance on text-based annotations, which fail to account for the distinct contributions of text, speech, and images (Du et al., 2023; 2024). Building on these insights, we introduce MuSeD, a Multimodal Sexism Detection dataset in Spanish with modality-specific labels, enabling a more granular evaluation of multimodal models.

# 3    MuSeD: A dataset for Multimodal Sexism Detection

MuSeD is a Multimodal Sexism Detection dataset in Spanish. We propose an annotation framework to analyze the impact of each modality (text, speech, and image) when identifying sexist content in videos.

## 3.1    Data sources

To ensure a balanced dataset that includes *sexist* and *non-sexist* content, we collected videos from two social media platforms: TikTok and BitChute. In February 2022, TikTok updated its community guidelines to explicitly ban misogyny and misgendering.[3] Despite these efforts, forms of denigration against women and people who do not conform to a cisgender identity persist on the platform in both explicit and implicit ways (Banet-Weiser & Maddocks, 2023). In contrast, BitChute is a low-moderation content platform that makes no effort to prevent forms of hate speech, including misogyny. It was launched in 2017 as an alternative to the heavily moderated platform YouTube. The key difference between YouTube and BitChute is that BitChute does not use a personalized recommendation algorithm. Instead, the platform suggests popular videos on the front page and related videos while the user watches content. Video selection is determined by the channel owner rather than a recommender system.

---

[3]https://www.tiktok.com/community-guidelines/en

## 3.2 Data collection

The data was collected between May and October 2024 from TikTok and BitChute. To gather videos from TikTok, we compiled a list of Spanish hashtags. We did not differentiate between European and Latin American Spanish and included both varieties. The list of Spanish hashtags includes 11 from Plaza et al. (2024), which we significantly expand with 176 additional hashtags to cover discrimination based on sexual orientation and gender identity. The original set is focused primarily on sexism against women, including terms such as #brecha salarial (gender pay gap), #violencia machista (gender-based violence), and #estereotipos de género (gender stereotypes). It also includes general terms such as "mujer" (woman) and "hombre" (man), which, while not explicitly linked to sexism, can be used in sexist contexts. We extended the set to include terms related to sexist content targeting sexual orientation and gender identity such as #lgbt, #binarie (binary), and #ideología de género (gender ideology). The final list consists of 187 hashtags encompassing various topics related to sexuality, gender, feminist discussions and discrimination based on sex and gender. Additionally, we identified a small number of prolific users (Anzovino et al., 2018) who frequently share sexist content—such as relationship advice and recommendations on how to be a "successful" man—as well as other users who share non-sexist content, including discussions on feminist topics and personal stories about experiences of sexist behavior. Based on this search, we collected video URLs associated with the selected hashtags using Apify (Arcos & Rosso, 2024).[4] We then collected the videos partially using Apify and partially through manual downloads. We ensured that all collected videos had been made publicly available by the creators. To obtain the videos from BitChute, we adapted a sample of hashtags from our curated list to use as keywords for the front page search. We then selected a set of videos from the retrieved list and collected them using the BitChutedl software.[5] The final dataset includes 400 videos, primarily from TikTok ($\approx$ 90%), while the remaining videos were collected from BitChute.

**Preprocessing**   We preprocessed the videos in three ways. First, we transcribed the audio into text using Whisper-CTranslate2, a Whisper implementation with diarization,[6] and we also included the timestamps. The automatic transcription was revised and corrected by a professional linguist. Second, we extracted the audio from the video using the command-line tool `FFmpeg`. Finally, we extracted text from frames using the Python module EasyOCR.[7]

## 3.3 Annotation process

**Mitigating annotation bias**   During the annotation process, various biases can arise due to multiple factors such as a lack of demographic diversity among annotators, preconceived stereotypes, and inadequate training (Hovy & Spruit, 2016; Sap et al., 2022). Following the methodology outlined by Zeinert et al. (2021), we employed multiple strategies to mitigate annotator biases.[8] First, we recruited annotators with diverse gender and age backgrounds.[9] The annotators were sourced through academic channels and all have a background in linguistics. One of the annotators is an expert in tasks related to the study topic. Second, the annotators underwent training prior to starting the annotation task, which included annotating a sample of 50 items followed by a discussion. Third, we held weekly meetings and, when necessary, updated the annotation guidelines. Each meeting was moderated by an expert on sexism on social media, who led the discussions, particularly on ambiguous cases in which sexism appeared in implicit forms. Although all of our annotators were European, we made concerted efforts to alleviate potential bias arising from limited demographic diversity. We ensured that they were familiar with the social and cultural contexts of Spanish-speaking regions, including both European and Latin American

---

[4] https://apify.com
[5] https://pypi.org/project/bitchute-dl/
[6] https://github.com/Softcatala/whisper-ctranslate2
[7] OCR with https://pypi.org/project/easyocr/
[8] See the Ethics statement in Section 5 for details.
[9] See Table 5 in Appendix B.

contexts. Annotators were expected to fully understand the content of each video, and in cases of dialectical complexity, these instances were collaboratively discussed to reach a shared understanding and maintain consistent annotations.

**Annotation framework**  In designing our annotation framework, we introduced a novel methodology to classify *sexist* and *non-sexist* content across different modalities. The annotation guidelines are included in Appendix A. Our annotation process consisted of three levels: first, annotators labeled the transcript and the OCR texts;[10] second, they annotated the audio; and finally, they annotated the entire video, which included all the modalities. OCR transcriptions were also considered as text because we observed that, in some cases, classification would not be possible without them due to the lack of contextual information. This was particularly relevant in 13 cases with only background music and no identifiable speakers. The group of annotators was split into two teams, each consisting of three people (two who identify as women and one who identifies as a man). Each team annotated half of the dataset in two formats—text and video—while the audio was annotated by the team that did not annotate the text and video. In all three steps, annotators classified the given item (text, audio, or video) as *sexist* or *non-sexist*, using Label Studio as a platform.[11] Moreover, for the video annotation task, they selected the segment they identified as containing sexism.

Content is considered sexist in four main cases. **Stereotype**: it defines a set of properties that *supposely* differentiate men and women, based on stereotypical belief (Samory et al., 2021); **Inequality**: it claims that gender inequalities no longer exist and that the feminist movement is marginalizing the position of men in society (Samory et al., 2021); **Discrimination**: it discriminates against individuals based on their sexual orientation or gender identity and denigrates the LGBTQ+ community (Chakravarthi et al., 2021), often by accusing them of promoting a "gender ideology"; **Objectification**: it depicts women as physical objects valued primarily for their utility to others (Szymanski et al., 2011). This type of content is more prevalent in visual form, as images intensify the hypersexualization of female bodies. Table 1 includes examples for each case of sexism. If an item contains a denunciation or a report of a sexist experience, it must be labeled as non-sexist (Chiril et al., 2020).

### 3.4  Dataset analysis

Our final dataset, MuSeD, includes 400 videos spanning over $\approx$ 11 hours of content.[12] We observe that the most frequent hashtag is *#feminismo* (feminism), followed by *#hombres* (men), *#mujeres* (women), *#machismo* (male dominance), and *#identidadsexual* (sexual identity).[13] The class distribution is balanced with 48.5% of videos labeled as *sexist* and 51.5% labeled as *non-sexist*.[14] We conduct a deeper analysis of the sexist videos by examining how many were categorized into the four cases of sexism described in Section 3.3. The majority of these videos were labeled as *Stereotype* (56.2%), followed by *Inequality* (39.2%), *Discrimination* (15.5%), and *Objectification* (3.6%).

**Inter-annotator agreement (IAA)**  The reliability of the annotations was evaluated by analyzing the level of agreement among annotators (Artstein, 2017). To measure IAA, we used Fleiss' kappa (Fleiss, 1971), observing a consistent increase in agreement when all modalities were included. The agreement increases from 0.719 (Team 1) and 0.717 (Team 2) for text annotation to 0.741 (Team 2) and 0.824 (Team 1) for audio. For video annotation, the IAA reaches 0.834 for Team 1 and 0.853 for Team 2, indicating an improvement from *substantial* to *almost perfect* agreement (Landis & Koch, 1977; Artstein & Poesio, 2008) when comparing text and video annotations. In comparison, the study by Arcos & Rosso (2024) reported an IAA of 0.499 for the task of sexism detection, indicating *moderate* agreement. Based on

---

[10] See preprocessing in Section 3.2.

[11] https://labelstud.io.

[12] See Table 6 in Appendix D for the dataset statistics.

[13] See Figure 3 for the distribution of video lengths and Figure 4 for the frequency of the top 10 hashtags in Appendix D.

[14] See Table 7 for the distribution of sexist and non-sexist videos on TikTok and BitChute.

discussions with the annotators, we identified two main challenges in recognizing sexism: the difficulty of identifying implicit forms of sexism and, in some cases, the contradiction between verbal and non-verbal content.[15]

**Evaluating annotation agreement on BitChute videos**    To evaluate the reliability of annotation agreement, we also selected a sub-sample of the dataset that includes videos extracted from BitChute. We expected the majority of these videos to be marked as sexist, given that BitChute is a platform with *low-moderation*. We found that, in the text annotation task, 87.88% of the videos were marked as sexist. This percentage increased in the video annotation task, where 93.94% of the videos were classified as sexist.

## 4    Automatic sexism detection

We evaluate a range of large language models (LLMs) on the task of sexism detection using our introduced dataset, MuSeD. The goal is to classify each video, as either *sexist* or *non-sexist*. As in the manual annotation process, we provide models with either (i) text-only input, i.e., the video transcript and OCR, or (ii) a combination of text and visual content. This allows us to investigate whether incorporating additional modalities beyond text enhances the automatic detection of sexist content in social media videos.

### 4.1    Models

**Large language models**    We evaluate both smaller and larger LLMs. Among smaller models, we experiment with Llama-3-8B-Instruct (Dubey et al., 2024), Qwen2.5-3B-Instruct (Yang et al., 2024), and Salamandra-7B-Instruct (Gonzalez-Agirre et al., 2025). Additionally, we assess larger open and proprietary models, including Llama-3-70B-Instruct, Gemini-2.0-Flash (Team et al., 2024), Qwen2.5-32B-Instruct, GPT-4o (Hurst et al., 2024), and Claude 3.7 Sonnet (Anthropic, 2024). This setup allows us to examine (i) whether text alone provides a sufficient signal for automatic sexism detection in social media videos and (ii) how the performance of smaller models compares to that of larger models.

**Vision & language models**    We evaluate three proprietary multimodal LLMs—GPT-4o, Gemini 2 Flash, and Claude 3.7 Sonnet—all capable of processing multiple images as input (see data preprocessing in Section 4.2). Additionally, we assess Gemini 2 Flash (Video) using raw video input, allowing the model to process the video directly without preprocessing.[16] Each model receives the text transcript and visual content, enabling a comparative analysis of their ability to detect sexism in multimodal data.

**Data processing**    To ensure models focus on video content rather than metadata, we remove all timestamps from the transcripts. For multimodal LLMs, we preprocess videos by extracting frames at evenly spaced intervals following Sánchez Villegas et al. (2025). To maintain computational efficiency while preserving key visual information, we cap video duration at 80 seconds, given that the median length is 75.4 seconds. Frames are allocated adaptively, with a minimum of 2 and a maximum of 10 frames per video. Additionally, we filter out black frames, commonly found in TikTok videos during transitions, by detecting low pixel intensity. In this way, we ensure that only meaningful visual content is retained.

### 4.2    Experimental setup

We evaluate all models in a zero-shot setting by prompting them with a yes/no question about whether the video exhibits sexist content, giving them either the transcript alone or both text and visual information. The system instruction and user prompt are provided in Spanish. Model responses, given in Spanish (*Sí* for "Yes" or *No* for "No"), are then mapped

---

[15]See Appendix C for examples of the challenging cases.

[16]Other models in our evaluation do not support direct video input and require frame-based preprocessing.

|                          | Text Label | Multimodal Label |
|--------------------------|------------|------------------|
| Random Baseline          | 50.00      | 50.00            |
| Majority Baseline        | 52.95      | 51.50            |
| **Smaller LLMs**         |            |                  |
| Llama-3-8B-Instruct      | 51.33      | 50.33            |
| Qwen2.5-3B-Instruct      | **61.33**  | **60.50**        |
| Salamandra-7b-instruct   | 59.50      | 59.33            |
| **Larger LLMs**          |            |                  |
| Llama-3-70B-Instruct     | 71.50      | 69.42            |
| Gemini-2.0-Flash         | 68.25      | 67.25            |
| Qwen2.5-32B-Instruct     | 76.50      | 74.50            |
| GPT-4o                   | **80.50**  | **79.50**        |
| Claude-3.7 Sonnet        | 76.50      | 75.00            |
| **Multimodal LLMs**      |            |                  |
| GPT-4o                   | **83.50**  | **82.00**        |
| Gemini-2.0-Flash         | 72.50      | 72.50            |
| Claude-3.7 Sonnet        | **82.50**  | **82.00**        |
| Gemini-2.0-Flash (Video) | 68.92      | 67.17            |

Table 3: Accuracy for sexism detection using LLMs and Multimodal LLMs. Models are evaluated against the text label (assigned by annotators based on the text transcript alone), and the multimodal label (assigned when both text and visual content were available to annotators). Best results in each group are in bold.

to the corresponding labels (*sexist* or *non-sexist*).[17] We also conducted additional experiments using an extended, definition-based prompt[18] that explicitly includes the definition of sexist content, as detailed in our guidelines in Appendix A.

**Evaluation metrics**   We evaluate model performance with accuracy, given that the dataset is balanced.[19] Each model is assessed against two types of ground truth labels: the **Text Label**, which is assigned by annotators based on the text transcript alone, and the **Multimodal Label**, which is assigned when both text and visual content were available to annotators. With 7% of cases differing between labels, this evaluation enables the analysis of whether text models can predict sexism as perceived in a full multimodal context and whether vision-and-language models rely on visual cues that change their predictions with respect to text-based content. Additionally, we include random and majority label baselines.

### 4.3   Results

**Do LLMs effectively detect sexism?**   The results in Table 3 suggest that models can detect sexism to some extent, with accuracy varying across different model sizes.[20] Larger models generally outperform smaller ones. The best-performing model in this setting (text label) is GPT-4o, achieving 80.50% accuracy, followed by Claude 3.7 Sonnet and Qwen2.5-32B-Instruct (76.50%). The gap between models suggests that sexism detection remains a challenging task, especially for models with fewer parameters where alternative approaches such as few-shot prompting or task-specific training may be necessary to enhance sexism detection without requiring larger models.

---

[17]Implementation details are included in Appendix E and the prompts in Appendix F.

[18]The extended prompts are included in Appendix F.2.

[19]We include F1 scores in Appendix E.

[20]Table 9 reports LLMs performance broken down by dataset source, including accuracy for both TikTok and BitChute, as well as the difference between them.

| Label
Prompt | Text
Basic | Text
Definition | Multimodal
Basic | Multimodal
Definition |
|---|---|---|---|---|
| Random | 50.00 | 50.00 | 50.00 | 50.00 |
| Llama-3-70B-Instruct | 71.50 | 83.25 | 69.42 | 82.25 |
| Gemini-2.0-Flash | 68.25 | 75.25 | 67.25 | 72.75 |
| GPT-4o | **80.50** | 82.50 | 79.50 | 81.00 |
| Claude-3.7 Sonnet | 76.50 | **84.25** | 75.00 | **83.75** |
| **Vision & Language** | | | | |
| Gemini-2.0-Flash (Video) | 68.92 | 77.25 | 67.17 | 75.75 |
| Gemini-2.0-Flash | 72.50 | 74.00 | 72.50 | 74.00 |
| GPT-4o | 83.50 | 86.50 | **82.00** | 85.00 |
| Claude-3.7 Sonnet | 82.50 | **87.75** | **82.00** | **86.75** |

Table 4: Accuracy comparison across models using basic (yes/no) and definition prompts, evaluated against text and multimodal labels. Best results in each group are in bold.

**When does visual content enhance sexism detection?** Comparing results from text-only inputs with text and visual content (Table 3 Text Label), we observe a performance increase in some models. GPT-4o achieves 83.50% accuracy, while Claude 3.7 Sonnet achieves 82.50%. However, not all models benefit equally from visual information. For instance, Gemini 2 Flash performs worse with raw video input (68.92%) compared to preprocessed frames (72.50%), suggesting that data-specific preprocessing techniques like our frame extraction method (see Section 4.2) may enhance the quality of visual cues. This variation highlights that while visual information can improve performance, its effectiveness depends on how well a model integrates and uses multimodal data.

**How do LLMs perform against multimodal annotations?** Models perform slightly worse when evaluated against the multimodal label compared to the text label, suggesting that incorporating visual context introduces additional complexity. This pattern holds across both text-only and multimodal models. For instance, GPT-4o achieves 83.5% accuracy on the text label but drops to 82% on the multimodal label. Similarly, Claude-3.7 Sonnet sees a decrease from 82.5% to 82%. These differences indicate that visual signals may alter human judgments in ways that are harder for current models—particularly those trained primarily on text—to fully capture. Nonetheless, multimodal models still perform competitively across both settings, and their strong performance on the text label (e.g., GPT-4o outperforming all text-only models by at least 3 percentage points) suggests they effectively integrate textual information. Overall, the results highlight that the multimodal label introduces a more challenging benchmark and underscore the importance of visual context in shaping perceptions of sexism.

**Do LLMs benefit from a more detailed definition of sexism?** As shown in Table 4, incorporating the definition consistently improves model performance across both text-only and multimodal settings. For the text label, models such as Llama 3 and Claude 3 show notable improvements, with accuracy gains from 71.50% to 83.25% and from 76.50% to 84.25%, respectively. Similarly, for the multimodal label, GPT-4o and Claude 3 achieve accuracy improvements from 82.0% to 85% and 86.75% respectively. These results show that providing a more detailed definition enhances not only human annotation quality but also the model's ability to accurately identify sexist content, underscoring the importance of precise task formulation.

**How important is text for sexism detection in videos without speech?** We also analyze the 13 cases where videos contain primarily background music, meaning the main textual information comes from OCR. GPT-4o achieves 84.6% accuracy on the text label when using text input alone, but drops to 54.8% on the multimodal label when incorporating both text and images. Similarly, Claude 3.7 Sonnet achieves 76.9% accuracy with text-only input and 61.5% with multimodal input. These results highlight the critical role of textual information, even when derived solely from OCR, in detecting sexism.

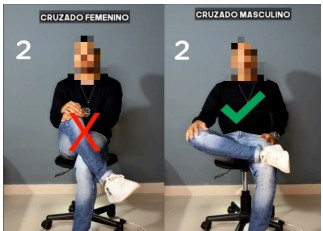

*Alpha. Piernas separadas. Cruzado masculino. El frente. Inclinarse hacia atrás. La cabeza. Brazos cómodos.*
**English translation:** *Alpha. Legs apart. Male crossing. The front. Leaning back. The head. Arms relaxed.*

Figure 1: Example of a video misclassified by GPT-4o. It contains implicit sexism, relying on cultural context and social norms the model struggles to interpret.

**How do LLMs perform in detecting sexism in videos from BitChute?** In the same way as for the annotation task, we expect the models to correctly classify a consistent set of videos collected from BitChute as sexist (see Section 3.4). GPT-4o reaches 88% accuracy on the text label when using only textual input, but its performance decreases to 82% on the multimodal label when both text and images are used. In contrast, Claude 3.7 Sonnet shows the opposite trend: it achieves 70% accuracy with text-only input, which rises to 94% when incorporating multimodal input. This aligns with Table 3, where Claude 3.7 Sonnet's performance improves substantially when provided with visual information.

**Do models struggle with the same cases as human annotators?** We analyze GPT-4o's misclassifications across three setups: *text-only* errors, *text & visual* errors, and consistent errors in both, using the multimodal label as it provides the most comprehensive annotation, where annotators had access to both text and visual information. Incorporating visual content reduces errors from 82 (*text-only*) to 72 (*text & visual*), indicating that multimodal information improves classification. However, 44 cases remain misclassified across both settings, suggesting inherent challenges in detecting certain instances. Similar to the challenges encountered during the annotation process (see Section 3.4), we find that 20% of misclassified cases involve implicit sexism, such as stereotypes. For example, Figure 1 shows a misclassified video where a man demonstrates two ways of sitting and asserts the "correct" one for men. This highlights how implicit sexism relies on cultural context and social norms (Frenda et al., 2019), which models struggle to interpret. The inter-annotator agreement for these cases in the video annotation task is lower (0.61/0.70) than for the full dataset (0.83/0.85), highlighting the challenge of identifying implicit sexism.

## 5 Conclusion and future work

In this work, we introduced MuSeD, the first multimodal dataset for sexism detection in Spanish social media videos, incorporating discrimination based on sex, sexual orientation, and gender identity. This is the first study to analyze data from both a moderated platform (TikTok) and a low-moderation platform (BitChute), providing insights into sexism detection across different content moderation environments. We introduced a new annotation framework that structures the process into distinct stages: text, audio, and full video, enabling a more granular analysis of multimodal content. We evaluated a range of LLMs and multimodal LLMs on the task of sexism detection using MuSeD. Our results show that multimodal information improves the classification of sexist content for both humans and models compared to using only text. Both struggle with implicit sexism, which depends on social and cultural factors. We also show that including a more detailed definition of sexism in the prompt improves the model accuracy for both text and multimodal labels compared to using a basic prompt, highlighting the importance of a clear task definition. Future studies could examine specific types of sexism, such as sexual orientation stereotypes (Morrison & Bearden, 2007).

## Acknowledgments

This work was supported by the 'FairTransNLP-LANGUAGE: Analyzing toxicity and stereotypes in language for unbiased, fair, and transparent systems' project (MCIN/AEI/10.13039/501100011033/FEDER,UE), and funded by the CLiC research group (2021SGR00313) and a research grant (VIL53122) from VILLUM FONDEN. We acknowledge EuroHPC Joint Undertaking for awarding us access to Karolina, hosted by IT4Innovations, Czech Republic.

## Ethics statement

The primary intent of this work is to advance research in the automatic detection of harmful content on social media platforms. Our goal is to contribute to the development of AI-driven moderation systems that help create a fair and safe online environment while also supporting human moderators, who are at risk of mental health issues due to prolonged exposure to harmful content. To ensure ethical and unbiased data annotation, we carefully recruited annotators with diverse gender and age backgrounds, minimizing annotation bias in the labeling process. All annotators provided informed consent to engage with potentially offensive material. Additionally, we conducted weekly discussions where annotators could openly share their thoughts and emotional responses, with a moderator ensuring a supportive and respectful environment. The annotators were paid €10 per hour (50 hours in total). To uphold ethical standards, we manually excluded videos containing explicit violent content and only collected publicly available videos shared by their creators. None of the authors directly participated in the annotation process.

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

# A   Annotation guidelines

The focus of the annotation is to classify content as either *sexist* or *non-sexist*. Content is considered sexist if it includes:

**Stereotype**   Stereotypes appear in two main forms. Firstly, they include the formulation of a set of descriptive properties that supposedly distinguish men and women based on gender stereotypes (1). Secondly, they include the formulation of a prescriptive set of qualities that men and women are expected to adhere to in order to fulfill gender roles in society (2).

(1) Me he encontrado con una chica que explica el por qué no le gusta a muchos hombres. Y yo pregunto a esta mujer, ¿tú estarías contigo en pareja? **¿Te imaginas convivir con una marimandona? Porque, ¿qué significa tener las cosas claras? Que tú quieres decorar el salón de rosa. Y como tienes las cosas muy claras y eres muy mandona, el salón se va a decorar de rosa. Los hombres no quieren estar contigo no porque no te puedan controlar, sino porque no te soportan**.
*English translation*: I came across a girl who explains why many men don't like her. And I ask this woman, would you be in a relationship with yourself? **Can you imagine living with a bossy woman? Because, what does it mean to have things clear? That you want to decorate the living room in pink. And since you have everything very clear and you're so bossy, the living room will be decorated in pink. Men don't want to be with you not because they can't control you, but because they can't stand you.**

(2) **¿Qué hace un alfa cuando una mujer lo rechaza? No le preguntes por qué**. Muchos hombres, cuando una mujer los rechaza, se lo toman muy personal y tratan de obtener una explicación de por qué los están rechazando. **Nunca le pidas una explicación, porque te hará ver como un hombre necesitado**.
*English translation*: What does an alpha do when a woman rejects him? Don't ask her why. Many men, when a woman rejects them, take it very personally and try to get an explanation for why they are being rejected. **Never ask for an explanation, because it will make you look like a needy man**.

**Inequality**   Content that denies the existence of inequalities (past or present) between men and women (3). They are also opposed to feminism, by claiming that the movement marginalizes men (4).

(3) **Es hora de admitirlo, los hombres tenemos privilegios por sobre las mujeres, sociales y legales** [..]. **Tenemos seis años más de pena por el mismo delito cometido por una mujer y el doble de posibilidades de ser encarcelado**.
*English translation*: It's time to admit it, **men have privileges over women, both socially and legally** [. . .]. **We have six more years of punishment for the same crime committed by a woman and twice the chances of being incarcerated.**

(4) **El feminismo te lavó el cerebro y te arruinó la vida. Porque hoy en día, si tú le preguntas a una mujer para qué necesitamos a los hombres, ellas dicen que para nada**. Y eso lo pueden comprobar en muchas entrevistas. **En una entrevista, una chica dijo que pudiéramos sobrevivir sin los hombres porque al cabo ya está todo construido. [. . . ] A los hombres sí los necesitamos, aunque tu mente delirante te diga que no**.
*English translation*: **Feminism washed your brain and ruined your life. Because nowadays, if you ask a woman why we need men, they say for nothing.** And this can be confirmed in many interviews. **In one interview, a girl said that we could survive without men because, after all, everything is already built. [. . . ] We do need men, even though your delirious mind tells you otherwise.**

**Discrimination**   Content that discriminates against individuals based on their sexual orientation and gender identity, including criticism of the ability to choose a gender different from the sex assigned at birth (5), as seen in the case of transgender individuals.

(5) La doctora McNamara afirma que sus puntos de vista se basan en la ciencia, pero afirmar que el sexo se asigna al nacer no tiene ninguna base científica en absoluto. **Su lenguaje engaña a la gente, especialmente a los niños, haciéndoles pensar que hombre y mujer son designaciones arbitrarias que pueden cambiar. Eso simplemente no es cierto**.
*English translation*: Dr. McNamara claims that her views are based on science, but stating that sex is assigned at birth has no scientific basis whatsoever. **Her language deceives people, especially children, making them think that man and woman are arbitrary designations that can change. That is simply not true.**

**Objectification**  Content that portrays women as objects, evaluating their physical appearance, and criticism for not conforming to normative beauty standards (6). This type of content is more commonly found in videos, where visual information plays a crucial role.

(6) Dicen que un hombre ama a una mujer mucho más de lo que una mujer ama a un hombre, y estoy empezando a creer que es cierto. ¿Por qué? Pues para empezar, un hombre ama a una mujer con muchas menos condiciones. **Lo único que un hombre pide para amar a una mujer es que sea femenina, atractiva y fértil**. Y a cambio, él da todo su esfuerzo, trabajo y recursos para esa mujer y su familia.
*English translation*: They say that a man loves a woman much more than a woman loves a man, and I'm starting to believe that it's true. Why? Well, to begin with, a man loves a woman with far fewer conditions. **The only thing a man asks to love a woman is that she is feminine, attractive, and fertile**. And in return, he gives all his effort, work, and resources to that woman and her family.

## B  Annotator profiles

Table 5 shows the annotator profiles. The annotators were recruited through academic channels and all have a background in linguistics. Two are current undergraduate students pursuing a degree in linguistics, one is an adjunct professor specializing in linguistics, and the remaining annotators were selected based on prior academic experience in data annotation and corpus creation. All annotators received comprehensive training before beginning the annotation process to ensure consistency and reliability.

| Annotator profiles | |
|---|---|
| Gender | 4 female, 2 male (6 total) |
| Age | 3 under 30, 3 over 30 |
| Native Language | European Spanish, Catalan |
| Study/Occupation | Linguistics consultant (1), Linguistics students (2), Teacher (2), Adjunct Professor in Linguistics (1) |

Table 5: Annotator profiles, including gender, age groups, languages, and areas of study/occupation.

## C  Challenging cases

Based on the discussions with the annotators, the following types of cases were the most challenging to annotate:

- **Implicit sexism.** The creator's intentions were ambiguous, and sexism did not appear explicitly but rather implicitly. Implicit forms of sexism were the most difficult to classify because disambiguating the creator's intentions was more challenging, as seen in the following case:

  (7) No estás mal de la cabeza, es que eres ecosexual [. . . ], es que tienes una orientación sexual diferente. Lo único que debes de hacer es respetar a las plantas, que siempre te den su consentimiento y no olvides llevar tu contrato de relaciones sexuales seguras.

*English translation*: You're not crazy, it's just that you're ecosexual [. . . ], it's just that you have a different sexual orientation. The only thing you need to do is respect the plants, always get their consent, and don't forget to carry your safe sex contract.

- **Contradictory information.** Other challenging cases involved contradictions between verbal and non-verbal content. Figure 2 illustrates an example of this type of case, where a user disapproves of what another person says through facial expressions.

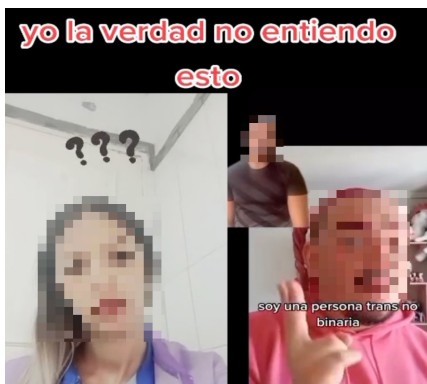

*Soy una persona trans, no binaria. Más específicamente soy demigirl. Mi orientación sexual es polisexual.*
**English translation:** *I am a trans, non-binary person. More specifically, I am a demigirl. My sexual orientation is polysexual.*

Figure 2: Example of a challenging case in the video annotation task, where verbal and non-verbal content contradict each other.

## D  Corpus statistics

Table 6 presents the dataset statistics, while Table 7 shows the distribution of sexist and non-sexist videos across TikTok and BitChute. Figure 3 shows the distribution of the video lengths, and Figure 4 shows the frequency of the top 10 hashtags used.

| Dataset statistics | |
| --- | --- |
| Number of videos (total) | 400 |
| Total length | 10h 50m 25s |
| Average length | 97.56s |
| Median length | 75.38 |
| Number of TikTok videos | 367 |
| Number of BitChute videos | 33 |
| Length videos TikTok | 9h 53m 25s |
| Length videos BitChute | 57m |

Table 6: Dataset statistics, including total, average, and median video lengths, as well as the distribution of video duration across TikTok and BitChute.

| Platform | Number of Videos | Sexist Cases | Non-Sexist Cases |
| --- | --- | --- | --- |
| TikTok | 367 | 164 | 203 |
| BitChute | 33 | 30 | 3 |

Table 7: Distribution of sexist and non-sexist videos on TikTok and BitChute.

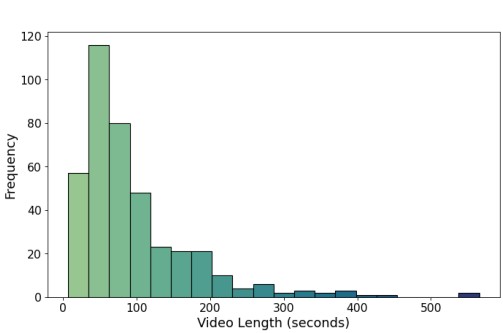

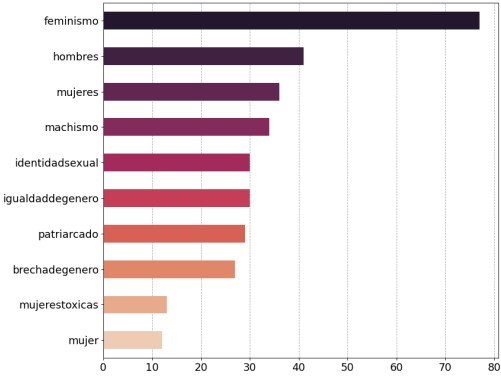

Figure 3: Length distribution. Distribution of video lengths in the dataset, highlighting a higher concentration of shorter videos and a long tail of longer videos.

Figure 4: Top 10 hashtags, with #*feminismo* being the most common, followed by #*hombres* and #*mujeres*, reflecting key themes in the collected videos.

## E    Implementation details and results

**Implementation details**    Experiments with Llama 3 and the Qwen2.5-32B-Instruct models are conducted on eight NVIDIA A100 GPUs, while the smaller LLMs, Qwen2.5-3B and Salamandra-7b were run on a single NVIDIA GeForce RTX 4090. Experiments for each model were run in under 10 minutes. For closed models, we use their respective APIs. The total cost was 54 USD.

**Results**    Table 8 shows the F1 scores for the models evaluated in the text label (assigned by annotators based on the text transcript alone), and the multimodal label (assigned when both text and visual content were available to annotators).

| Model | Text Label | Multimodal Label |
|---|---|---|
| Random Baseline | 53.60 | 49.88 |
| Majority Baseline | 68.85 | 67.99 |
| **Smaller LLMs** | | |
| Llama-3-8B-Instruct | 35.62 | 34.33 |
| Qwen2.5-3B-Instruct | **60.80** | **59.40** |
| Salamandra-7b-instruct | 58.60 | 58.95 |
| **Larger LLMs** | | |
| Llama-3-70B-Instruct | 68.60 | 67.72 |
| Gemini-2.0-Flash | 68.25 | 67.24 |
| Qwen2.5-32B-Instruct | 76.60 | 74.50 |
| GPT-4o | **80.46** | **79.45** |
| Claude 3.7 Sonnet | 76.34 | 74.82 |
| **Multimodal LLMs** | | |
| GPT-4o | **83.33** | 81.80 |
| Gemini-2.0-Flash | 72.42 | 72.41 |
| Claude-3.7 Sonnet | 82.51 | **82.00** |
| Gemini-2.0-Flash (Video) | 68.93 | 67.16 |

Table 8: F1 scores for sexism detection using LLMs and Multimodal LLMs. Models are evaluated against the text label (assigned by annotators based on the text transcript alone), and the multimodal label (assigned when both text and visual content were available to annotators). Best results in each group are in bold.

Table 9 reports LLMs and Multimodal LLMs accuracy by dataset source. Smaller models tend to exhibit a wider performance gap between the two data sources, while larger models show more consistent performance.

| Model | BitChute Accuracy | TikTok Accuracy | Difference |
|---|---|---|---|
| **LLMs (Text Label)** | | | |
| Llama-3-8B-Instruct | 0.12 | 0.55 | 0.43 |
| Qwen2.5-3B-Instruct | 0.42 | 0.63 | 0.21 |
| Salamandra-7b-instruct | 0.70 | 0.58 | -0.12 |
| Llama-3-70B-Instruct | 0.42 | 0.74 | 0.32 |
| Gemini-2.0-Flash | 0.64 | 0.69 | 0.05 |
| Qwen2.5-32B-Instruct | 0.70 | 0.77 | 0.07 |
| GPT-4o | 0.88 | 0.80 | -0.08 |
| Claude 3.7 Sonnet | 0.70 | 0.77 | 0.07 |
| **Multimodal LLMs (Multimodal Label)** | | | |
| GPT-4o | 0.82 | 0.82 | 0.00 |
| Gemini-2.0-Flash | 0.79 | 0.72 | -0.07 |
| Claude 3.7 Sonnet | 0.94 | 0.81 | -0.13 |
| Gemini-2.0-Flash (Video) | 0.70 | 0.67 | -0.03 |

Table 9: Accuracy of LLMs and Multimodal LLMs for sexism detection across BitChute and TikTok.

# F Evaluation prompts

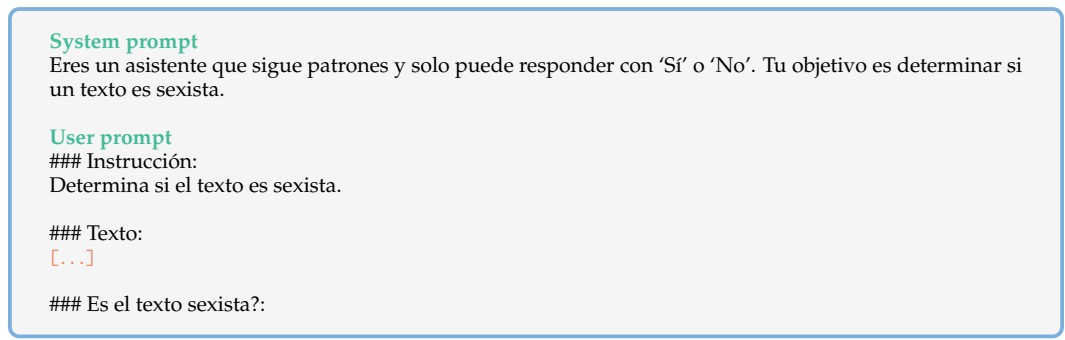

**System prompt**
Eres un asistente que sigue patrones y solo puede responder con 'Sí' o 'No'. Tu objetivo es determinar si un texto es sexista.

**User prompt**
### Instrucción:
Determina si el texto es sexista.

### Texto:
[...]

### Es el texto sexista?:

Figure 5: Prompt used to classify text transcripts from videos using text-only models. The placeholder [...] indicates where the text to be classified is inserted.

---

**System prompt**
Eres un asistente de clasificación de videos. Responde solo con 'Sí' o 'No'.

**User prompt**
### Instrucción:
Analiza el contenido de este video y determina si es sexista. Si el texto del video es sexista, el video también debe clasificarse como sexista, incluso si las imágenes no parecen ofensivas.

### Texto:
[...]

### Pregunta:
Basado en el texto y las imágenes proporcionadas, ¿el video es sexista? Responde solo con 'Sí' o 'No'.
Si el texto es sexista, responde 'Sí' sin importar las imágenes.

---

Figure 6: Prompt used to classify videos using Multimodal LLMs. The placeholder [...] indicates where the text to be classified is inserted. The images are also given to the models.

---

**System prompt**
Eres un asistente de clasificación de videos. Responde solo con 'Sí' o 'No'.

**User prompt**
### Instrucción:
Analiza el contenido de este video y determina si es sexista. Si el texto del video es sexista, el video también debe clasificarse como sexista, incluso si el video no parece ofensivo.

### Texto:
[...]

### Pregunta:
Basado en el texto y el video proporcionado, ¿el video es sexista? Responde solo con 'Sí' o 'No'.
Si el texto es sexista, responde 'Sí' sin importar el video.

---

Figure 7: Prompt used to classify videos using Gemini-2.0=Flash (Video). The placeholder [...] indicates where the text to be classified is inserted. The video is also given to the model.

## F.1 Alternative prompts

In the case of Qwen2.5-32B-Instruct, we find the best results using an analogous text prompt translated to English. The text label accuracy increase to 79.13% and to 78.36% in the case of the multimodal label. This great improvement is likely due to the model train set containing more English language tokens.

---

**System prompt**
You are a pattern-following assistant that can only answer with "Yes" or "No". Your goal is to determine whether a text is sexist.

**User prompt**
### Instruction:
Determine if the text is sexist.

### Text:
[...]

### Is the text sexist?:

---

Figure 8: English prompt used to classify text transcripts from videos using text-only models. The positions are indicated by placeholders [...], where the text to be classified is inserted.

## F.2 Extended prompts

**System prompt**
Eres un asistente de clasificación de videos. Responde solo con 'Sí' o 'No'.

**User prompt**
### Instrucción:
Determina si el texto es sexista. A continuación se presentan criterios que definen el sexismo:

- **Estereotipos:**
    - (1) Formulación de propiedades descriptivas que supuestamente distinguen a hombres y mujeres basadas en estereotipos de género.
    - (2) Formulación de propiedades prescriptivas que hombres y mujeres deben cumplir para encajar en los roles de género establecidos por la sociedad.
- **Desigualdad:**
    - (3) Contenido que niega la existencia de desigualdades (pasadas o presentes) entre hombres y mujeres.
    - (4) Contenido que se opone al feminismo, argumentando que este movimiento margina a los hombres.
- **Discriminación:**
    - (5) Contenido que discrimina a personas por su orientación sexual (por ejemplo, personas homosexuales, lesbianas, intersexuales, bisexuales, asexuales y pansexuales), por su identidad de género, incluyendo críticas hacia quienes eligen un género distinto al asignado al nacer (por ejemplo, personas transgénero). También contenido que discrimina a personas LGBTQ+.
- **Cosificación:**
    - (6) Contenido que presenta a las mujeres como objetos, evalúa su apariencia física o las critica por no ajustarse a estándares normativos de belleza.

### Text:
[. . .]

### ¿Es el texto sexista?:

Figure 9: Extended prompt used to classify text transcripts from videos using text-only models. The placeholder [. . .] indicates where the text to be classified is inserted.

