# OpenReview forum: "MuSeD: A Multimodal Spanish Dataset for Sexism Detection in Social Media Videos"
_colmweb.org/COLM/2025/Conference — COLM 2025_

### Official Review · Reviewer_4cEJ · 2025-05-12

**Rating:** 6
**Confidence:** 4
**Ethics Flag:** 1

**Summary:**

The paper is divided into two parts. In the first part, the authors introduce MuSeD (for Multimodal Spanish dataset for Sexism Detection), a multimodal dataset focused on Sexism Detection. In the second part, the authors use this dataset to evaluate the performance of several LLMs in this task.

Regarding MuSeD, the authors claim that it is the first dataset of its type. It is formed by a manually-selected and manually-tagged balanced set of sexist and non-sexist Spanish-language (both Latin-American and Iberian Spanish) videos. These videos, all publicly-available, where obtained from TikTok and BitChute platforms. While TikTok has a stronger moderation policy, BitChute is the opposite (thus, most BitChute videos contained in the dataset are sexist). The definition of sexism used here is a extended definition to encompass differetnt types of discrimination based on sex, sexual orientation and gender identity corresponding to 4 subcategories: stereotype, inequality, discrimination and objectification. In contrast, previous work focused just focused in sexism based in sex.

Previous works in multimodal sexism detection were focused in memes (i.e. images), while the new work goes beyond and focuse on videos. In contrast, related multimodal previous works using videos are more focused in hate speech instead of sexism.

All the dataset creation process is  explained in detail, also proposing an innovative anotation framework, with annotations conducted at different levels (both images themselves, OCRed text shown with the images, and text transcriptions of the audio). They also include a finer-grain annotation by identifying, not only wether the video is sexist or not, but also identifying the specific segment of the video containing such content. The resulting dataset, of great quality, contains 400 videos apanning oover 11h, with a balanced set of sexist and non-sexist items.

Regarding the second part of the paper, the authors use their dataset to evaluate the performance of several LLMs when facing a sexism detection task (binary classification: sexist or non-sexist). They several LLMs covering different types and sizes, both open and propietary models and both text-only models and multimodal models using audio transcriptions, OCRed images, video frames or videos as input. The models are evaluated in a zer-shot setting by prompting them with a yes/no question about whether the video contains sexist content. The results are analyzed using Accuracy as metric, obtaining positive results, specially with larger models. The results also demonstrate that a multimodal analysis can improve performance, but it also depend on how the relevent information is integrated in the process. However, LLMs also have problems when dealing with specially hard cases of implicit sexism the same as human annotators.

**Questions To Authors:**

* I think a mere "is this sexist?" prompt (in practice it's just that) is not enough here. Beware that the authors themselves state at the beginning that they are using a "broaded" definition of sexism. We don't know which "definition" is using the LLM. We should, at least, extend the evaluation experiments by using a second prompt inclusing the definition of sexism to be used. A few-shot approach is also desirable.

* Sect.1, l.54-73: Research questions and contributions should appear separately, and they should be itemized to improve reading.
* Sect.3.2, l.172-173: Why do you need translation (or do you mean "transcription")? Explain.
* Sect.3.4, l.229-234: Due to the nature of the platform, most BitChute videos contain sexist content. So, if you are able to identify (in some way) that your input video comes from BitChute, your best bet is to tag it as "sexist". Are you sure that there is no way to identify a video as a TikTok video or a BitChute video? That would adulterate the experiment.
* Sect.4.1, l.251-256: The authors should explain before reaching this paragraph (not later as it is now), that the video will be captured to get frames, etc. It will help the reader to understand what happens with the multimodal image-based LLMs.
* Sect.4.3, l.297-306: The question made is somewhat tricky. We are working with videos, not text tweets. If the content is transmitted using a video, it's because the message also uses images to improve transmission. So, we should expect that multimodal analysis improve sexism detection when dealing with videos with sexist content.
* Sect.4.3, l.307-313: Similarly, if a video contains text in the images (specially if it has no speech), it is because that text is used to transmit a message. Thus, text-models should not be completely useless here but, of course, the image keeps being of key importance (again, it's a video, not a text tweet).
* Sect.4.3, l.314-322: This paragraph has no interest. The source of the video is not important for this research work.
* Sect.5, l.340-341: The same when the authors say that "This is the first study to analyze data from both a moderated platform (TikTok) and a low-moderation platform (BitChute)". It has no interst.
* References: some of them conyain the usual lowercased errors

**Reasons To Accept:**

* Very well written. Very well organized. Easy to read.
* Multimodal analysis is a hot topic nowadays.
* The first dataset of its type (according to the authors), also defining an innovative annotation framework.
* They evaluate and compare both text-only models and multimodal models of different types for the task.

**Reasons To Reject:**

* Although very meritable and useful (referring to the dataset), it's the classic work introducing another dataset and trying different state-of-art approaches with it. No breakthrough.
* I would have expected further evaluation experiments with more elaborated, less naive prompts.

---

> ### Author Response · Authors · 2025-06-02
> **Responses to Reviewer 4cEJ**
>
> We thank Reviewer 4cEJ for highlighting the key strengths of our work. We're glad you found the paper clear and readable, and that you valued the focus on multimodality and the introduction of an innovative annotation framework. We also appreciate your acknowledgment of our experiments comparing text-only and state-of-the-art multimodal models for the task of sexism detection.
>
> **W1: Response to Concern Regarding Novelty.** Our work offers important contributions beyond the introduction of a new dataset:
>
> - Prior research has focused on misogyny detection in memes or sexism targeting women; instead, MuSeD addresses subtle, multimodal expressions of sexism in videos and adopts a broader definition covering sex, sexual orientation, and gender identity.
> - It is the first dataset with three-level annotations—text, audio, and video—enabling detailed modality analysis.
> - Combined with zero-shot evaluations of state-of-the-art models, MuSeD provides new insights into detecting nuanced, multimodal discrimination.
>
> **W2: Extended Prompt Evaluation with Definition of Sexism.** To address the reviewer’s concern regarding the use of only a basic prompt, we conducted additional experiments using an extended definition-based prompt that explicitly includes the definition of sexist content (as detailed in our guidelines in Appendix A). The extended prompts are as follows:
>
> **Instrucción:**
>    Determina si el texto es sexista.
>    A continuación se presentan criterios que definen el sexismo:
>
>    **Estereotipos:**
>
>    (1) Formulación de propiedades descriptivas que supuestamente distinguen a hombres y mujeres basadas en estereotipos de género.
>
>    (2) Formulación de propiedades prescriptivas que hombres y mujeres deben cumplir para encajar en los roles de género establecidos por la sociedad.
>
>    **Desigualdad:**
>
>    (3) Contenido que niega la existencia de desigualdades (pasadas o presentes) entre hombres y mujeres.
>
>    (4) Contenido que se opone al feminismo, argumentando que este movimiento margina a los hombres.
>
>    **Discriminación:**
>
>    (5) Contenido que discrimina a personas por su orientación sexual (por ejemplo, personas homosexuales, lesbianas, intersexuales, bisexuales, asexuales y pansexuales), por su identidad de género, incluyendo críticas hacia quienes eligen un género distinto al asignado al nacer (por ejemplo, personas transgénero). También contenido que discrimina a personas LGBTQ+.
>
>    **Cosificación:**
>
>    (6) Contenido que presenta a las mujeres como objetos, evalúa su apariencia física o las critica por no ajustarse a estándares normativos de belleza.
>
> **Texto:**
> [TEXT]
>
> ¿Es el texto sexista?
>
> As shown in the results, incorporating the definition consistently improves model performance across both text-only and multimodal settings. For the text label, models such as Llama 3 and Claude 3 show notable improvements, with accuracy gains from 70.75% to 83.25% and from 76.5% to 84.25%, respectively. Similarly, for the multimodal label, GPT-4o and Claude 3 achieve accuracy improvements from 82% to 85% and from 82% to 86.75%. These results show that providing a more detailed definition enhances not only human annotation quality but also the model’s ability to accurately identify sexist content, underscoring the importance of precise task formulation. Regarding the few-shot suggestion, we intentionally focus on a zero-shot setup to evaluate models' out-of-the-box capabilities without task-specific examples, aligning with realistic deployment scenarios.
> | Model                  | Text Basic | Text Definition | Multimodal Basic | Multimodal Definition |
> |-------------------------|------------|-----------------|------------------|-----------------------|
> | Random                  | 50.0%      | 50.0%           | 50.0%            | 50.0%                 |
> | Llama 3 70B Instruct     | 70.75%     | 83.25%          | 70.00%           | 82.25%                |
> | gemini-2.0-flash         | 68.25%     | 75.25%          | 67.25%           | 72.75%                |
> | GPT-4o                   | **80.50%** | 82.50%          | 79.50%           | 81.00%                |
> | claude-3-7-sonnet        | 76.50%     | **84.25%**      | 75.00%           | **83.75%**            |
> |                         |            |                 |                  |                       |
> | **Vision & Language**    |            |                 |                  |                       |
> | gemini-2.0-flash (video) | 68.92%     | 77.25%          | 67.17%           | 75.75%                |
> | gemini-2.0-flash         | 72.50%     | 74.00%          | 72.50%           | 74.00%                |
> | GPT-4o                   | 83.50%     | 86.50%          | **82.00%**       | 85.00%                |
> | claude-3-7-sonnet        | 82.50%     | **87.75%**      | **82.00%**       | **86.75%**            |

---

> > ### Author Response · Authors · 2025-06-02
> > **English translations of the prompts**
> >
> > We report here the English translations of the prompts, which explicitly include the definition of sexist content:
> >
> > **Instruction:**
> > Determine whether the text is sexist.
> > Below are the criteria that define sexism:
> >
> > **Stereotypes:**
> >
> > (1) Statements of descriptive properties that supposedly distinguish men and women based on gender stereotypes.
> >
> > (2) Statements of prescriptive properties that men and women must fulfill to fit into the gender roles established by society.
> >
> > **Inequality:**
> >
> > (3) Content that denies the existence of inequalities (past or present) between men and women.
> >
> > (4) Content that opposes feminism, arguing that this movement marginalizes men.
> >
> > **Discrimination:**
> >
> > (5) Content that discriminates against people based on their sexual orientation (e.g., homosexual, lesbian, intersex, bisexual, asexual, and pansexual individuals), their gender identity, including criticism of those who choose a gender different from the one assigned at birth (e.g., transgender people). Also, content that discriminates against LGBTQ+ individuals.
> >
> > **Objectification:**
> >
> > (6) Content that presents women as objects, evaluates their physical appearance, or criticizes them for not conforming to normative beauty standards.
> >
> > **Text:**
> > \[TEXT]
> >
> > Is the text sexist?

---

> > ### Author Response · Authors · 2025-06-02
> > **Responses to Reviewer 4cEJ**
> >
> > **Questions:**
> >
> > **​​Clarification on Readability Improvements.** Thank you for the comment. We will address this in the revised version of the paper.
> >
> > **Clarification on Translation vs. Transcription.** Thank you for the clarification. We meant “transcription,” not “translation”. This has been corrected on the paper.
> >
> > **Clarification on Platform Identification.** We appreciate this insightful question. We confirm that the models are unable to identify whether a video originates from TikTok or BitChute, as all metadata was removed during preprocessing. This prevents the models from having the information on the video’s source and mitigates any potential bias related to platform recognition.
> >
> > **Clarification on the Phrasing of the Question.** Thank you. We acknowledge that the phrasing of the original question may be unclear. In this analysis, we aim to study how LLMs perform on the task of sexism detection, which, as you mentioned, involves messages conveyed through multiple modalities. Our quantitative analysis shows that relying on text-only models is a limited approach, as they do not fully capture the multimodal content. We have revised the question to: “How do LLMs perform against multimodal annotations?”—a phrasing we believe better reflects the intent of the analysis.
> >
> > **Clarification on the Importance of Data Sources.** Thank you for the clarification. We sampled from different data sources to capture both implicit and explicit forms of sexism. For example, platforms like TikTok, which are moderated, tend to contain more implicit expressions of sexism, whereas platforms like BitChute—where there is little to no moderation—allow for more explicit hate speech. Additionally, as highlighted in related studies such as "Annotating Online Misogyny" (Zeinert et al., 2021), it is important to sample from multiple platforms to ensure data diversity, as different social media platforms attract different user bases and exhibit distinct domain-specific language.
> >
> > **Clarification on Data Preprocessing.** Thank you for the comment. We will address it in the revised version of the paper by inserting the description of the data preprocessing steps before the evaluation experiment section.
> >
> > **Lowercase Errors in References.** Thank you for the observation. We will correct it in the revised version of the paper.

---

> > > ### Comment · Reviewer_4cEJ · 2025-06-05
> > >
> > > Thanks for your responses. The new figures, obtained by providing the system with the proper definition of "sexism", are interesting. The authors should include them in the final version of the paper, if accepted.
> > >
> > > I have decided to keep my score. The paper has improved, indeed, but if my previous rating was 6 (but actually more like a 5.5), it keeps being 6 (but now so close to 7 to improve it).
> > >
> > > As I explained before, the work is meritable. If not accepted here, I encourage the authors to try in journals or conferences more focused on resources.

---

> > > > ### Author Response · Authors · 2025-06-10
> > > >
> > > > Thank you for the feedback. We're pleased to hear that you appreciate our additional experiments. We will incorporate the obtained results into the camera-ready version.

---

### Official Review · Reviewer_qK5A · 2025-05-13

**Rating:** 7
**Confidence:** 5
**Ethics Flag:** 2

**Summary:**

The paper introduces a new dataset for multimodal sexism detection in spanish language, namely MuSeD. The authors collected videos from social media platforms which underwent stringent quality assurance during the annotation process. MuSeD dataset offers both text only and multimodal labels for sexism detection for text and videos in spanish language.

**Ethics Concerns Details:**

Dataset contains samples with discrimination/stereotype/inequality/objectification in videos, especially sensitive samples from BitChute platform where annotators returned ~94% samples as sexist.

**Questions To Authors:**

1. Do you plan to make the dataset open source?
2. To alleviate data distribution bias, please include performance break up per dataset in multimodal benchmarks in Table 3 i.e what is the model performance on TikTok vs BitChute data.
3. Any analysis done on dataset break up with respect to different sexism cases as listed in Table 1 i.e how many samples from each of the a, b, c, d cases?
4. How are the annotators sourced (i.e annotation company, crowd sourcing, academia)? Did any of the authors participate in directly creating the annotations? Details unclear in appendix B.

**Reasons To Accept:**

- Introduction of a new multimodal dataset for online sexism detection in spanish language.
- Stringent quality assurance with respect to data collection, annotation and inter-annotator agreement.
- Benchmarking state of the art (SOTA) multimodal models on MuSeD dataset.

**Reasons To Reject:**

- Stronger motivation needed to argue why a new dataset in this domain of work i.e what data samples does MuSeD dataset cover not present in previous literature.
- Potential data distribution bias as details on data collection imbalance is vague i.e section 3.4 points out 48.5\% of the videos in MuSeD is sexist, of those 93.94\% of BitChute videos are labeled as sexist indicating majority of the sexist samples could be from the BitChute platform.
- possible annotation bias as all annotators are European as suggested in Appendix B whereas dataset contains samples covering both European and Latin American spanish. More discussion needed on how dialects are taken care of within the spanish language (e.g what dialects are covered/not covered in annotation guidelines?)

---

> ### Author Response · Authors · 2025-06-02
> **Responses to Reviewer qK5A**
>
> We thank Reviewer qK5A for the positive feedback. We’re pleased to hear that you appreciate the introduction of a new multimodal dataset for sexism detection in Spanish, the quality of our data collection and annotation methods, the computation of inter-annotator agreement, and the comparison of MuSeD to the state-of-the-art multimodal dataset for sexism and misogyny detection.
>
> **W1: Motivation for a new dataset in this domain.** MuSeD is the first dataset for sexism detection that introduces all of the following novel contributions:
>
> - The adoption of a broad definition of sexism that includes discrimination based on sex, sexual orientation, and gender identity.
> - The incorporation of data from two contrasting platforms—TikTok (moderated) and BitChute (low-moderation)—enabling the analysis of both subtle and overt forms of sexism.
> - The introduction of a three-level annotation framework (text, audio, and video), enabling detailed multimodal analysis of how each modality contributes to the detection of sexist content.
>
> **W2: Potential data distribution bias.** To address the reviewer’s concern regarding potential data distribution bias, we conducted a deeper analysis of the distribution of sexist and non-sexist videos on TikTok and BitChute. We found that the majority of videos from BitChute were labeled as sexist, while the majority of videos from TikTok were labeled as non-sexist. This confirms our expectation, given that BitChute is a low-moderation platform, whereas TikTok employs content moderation, and sexism tends to appear in less explicit forms compared to BitChute.
>
> | Platform  | Number of Videos | Sexist Cases | Non-Sexist Cases |
> |-----------|------------------|---------------|-------------------|
> | TikTok    | 367              | 164           | 203               |
> | BitChute  | 33               | 30            | 3                 |
>
> **W3: Potential annotation bias due to the annotators' background.** To mitigate potential annotation bias due to all annotators being European, we adopted several strategies. Our annotation team was diverse in terms of gender and age, and all members were familiar with the social and cultural contexts of Spanish-speaking regions, including both European and Latin American. Additionally, we ensured that annotators fully understood the content of each video; in cases of ambiguity or dialectal complexity, the team engaged in weekly discussions to reach a shared understanding and ensure consistent annotations.  Although annotators were based in Europe, we made concerted efforts to address dialectal variation collaboratively, ensuring that the dataset captures the diversity of Spanish dialects present in the videos.
>
> **Questions:**
>
> **1. Dataset Availability.** Yes, we plan to make the dataset available for research purposes. It will be published on the Zenodo platform, with the following access conditions: users must not share the dataset with anyone who was not part of the original request, and they must properly cite the paper in any publication, regardless of format or type.
>
> **2. Performance Breakdown by Dataset Source.** We identified a minor error in our script for the classification of videos from BitChute. Below, we report the corrected results. Smaller models tend to exhibit a wider performance gap between the two data sources, while larger models show more consistent performance. However, there is no clear pattern indicating which data source achieves higher accuracy.
>
> | LLMs (Text Label)                                   |
> |----------------------------------------------------|
>
> | Model                   | BitChute Accuracy | TikTok Accuracy | Difference |
> |-------------------------|-------------------|-----------------|------------|
> | Llama-3-8B-Instruct     | 0.12              | 0.55            | 0.43       |
> | Qwen2.5-3B-Instruct     | 0.42              | 0.63            | 0.21       |
> | Salamandra-7b-instruct  | 0.70              | 0.58            | -0.12      |
> | Llama-3-70B-Instruct    | 0.42              | 0.74            | 0.32       |
> | Gemini-2.0-Flash        | 0.64              | 0.69            | 0.05       |
> | Qwen2.5-32B-Instruct    | 0.70              | 0.77            | 0.07       |
> | GPT-4o                  | 0.88              | 0.80            | -0.08      |
> | Claude 3.7 Sonnet  | 0.70              | 0.77            | 0.07       |
>
>
>
> | Multimodal LLMs (Multimodal Label)                 |
> |----------------------------------------------------|
>
> | Model                     | BitChute Accuracy | TikTok Accuracy | Difference |
> |--------------------------|-------------------|-----------------|------------|
> | GPT-4o                   | 0.82              | 0.82            | 0.00       |
> | Gemini-2.0-Flash         | 0.79              | 0.72            | -0.07      |
> | Claude 3.7 Sonnet        | 0.94              | 0.81            | -0.13      |
> | Gemini-2.0-Flash Video   | 0.70              | 0.67            | -0.03      |

---

> > ### Author Response · Authors · 2025-06-02
> > **Responses to Reviewer qK5A**
> >
> > **3. Dataset Breakdown by Sexism Categories.** Thanks for the observation. We conducted a separate analysis on the 194 sexist videos, examining in detail how many were labeled according to the four types of sexism: Stereotype, Inequality, Discrimination, and Objectification. To perform this analysis, we used a majority vote approach: if at least two annotators assigned a specific label (e.g., "Stereotype") to a video, then that video was considered to include that type of sexism. Annotators were allowed to select one or more types of sexism per video. For example, a single video could contain a gender stereotype (Stereotype) and also deny the existence of gender inequality (Inequality). As shown in the table below, the majority of videos were labeled as Stereotype (56.2%), followed by Inequality (39.2%). Future work could focus on a deeper analysis of fine-grained sexism classification by both human annotators and models, using the MuSeD dataset.
> >
> > | Type of Sexism       | Count | Percentage (%) |
> > |------------------|--------|----------------|
> > | Inequality       | 76     | 39.2           |
> > | Discrimination   | 30     | 15.5           |
> > | Stereotype       | 109    | 56.2           |
> > | Objectification  | 7      | 3.6            |
> >
> > **4. Details on annotators.** None of the authors directly participated in the annotation process. The annotators were recruited through academic channels and all have a background in linguistics. With reference to Table 4 in Appendix B, two are current linguistics Bachelor’s degree students, one is an Adjunct Professor specializing in linguistics, and the remaining annotators were selected based on prior academic experience in data annotation and corpus creation. All annotators received comprehensive training before beginning the annotation process to ensure consistency and reliability.

---

> > > ### Comment · Reviewer_qK5A · 2025-06-02
> > > **Rating updated from 5 -> 7 subject to comments incorporated in draft**
> > >
> > > Please incorporate details mentioned in the comments in the main draft/appendix. Score updated from 5->7 subject to adding these details in camera ready paper in case of acceptance. Good luck!

---

> > > > ### Author Response · Authors · 2025-06-04
> > > >
> > > > Thank you for the positive feedback. We're glad you found our responses satisfactory, and we'll incorporate these changes into the camera-ready version.

---

### Official Review · Reviewer_xWDg · 2025-05-23

**Rating:** 6
**Confidence:** 4
**Ethics Flag:** 2

**Summary:**

This paper introduces MuSeD, a multimodal Spanish dataset for sexism detection in videos. It explores creating a systematic multimodal annotation framework, the role of visual information in identifying sexism, and how model misclassifications relate to human disagreement.

**Ethics Concerns Details:**

The proposed Spanish dataset for sexism detection may contribute to the emergence of sexual harassment-related public opinion and provide them with a channel.

**Questions To Authors:**

After reading the rebuttal, the authors have addressed all my confusion. Therefore, I would like to raise my score from to 6,

**Reasons To Accept:**

1. The introduction of MuSeD is a significant contribution. It addresses a growing need for resources to study multimodal sexism, particularly in Spanish, a widely spoken language. The dataset's inclusion of videos from both a moderated platform (TikTok) and a low-moderation platform (BitChute) allows for the study of diverse manifestations of sexism

2. The work expands the typical definition of sexism in such datasets beyond targeting only women, to include discrimination based on sexual orientation and gender identity. This is a commendable and timely extension.

**Reasons To Reject:**

1.  While 400 videos (≈11 hours) is a good starting point for a novel dataset and sufficient for the presented zero-shot evaluations, it may be relatively small for training more complex supervised multimodal models from scratch or for extensive fine-tuning. The authors might consider discussing plans or challenges for future expansion.

2. The annotation framework distinctly annotates text, audio, and then the full video. IAA for audio is reported separately. However, the experimental setup for models seems to primarily differentiate between "text-only" (transcript + OCR) and "text + visual" input. It's not entirely clear if or how non-textual audio features (e.g., tone, prosody, background sounds if not speech) were explicitly used by the multimodal models beyond the information contained in the human-corrected transcripts. If only transcripts were used, the "audio" annotation level's impact on models isn't directly assessed in the experiments.

3. The phrasing in the abstract and contributions could be slightly refined. The framework seems to analyze the contribution of modalities (textual, vocal, visual elements) towards arriving at labels (sexist/non-sexist), rather than analyzing "textual and multimodal labels" themselves. Labels are derived from assessing content through different modal lenses. This is a minor point of phrasing clarity.

4. While the inclusion of BitChute is a strength for diversity, the number of videos from BitChute is relatively small (33 videos, ≈10% of the dataset). While the high percentage of sexist content found aligns with expectations, the generalizability of model performance on this specific subset (e.g., Claude 3.7 Sonnet achieving 93.94% with multimodal input ) should be interpreted with caution due to the small sample size from this platform.

---

> ### Author Response · Authors · 2025-06-02
> **Responses to Reviewer xWDg**
>
> We thank Reviewer xWDg for the feedback. However, we are unclear why the perceived weaknesses outweigh the acknowledged strengths. In particular, the reviewer recognized our dataset as a significant contribution to multimodal sexism research and appreciated our broader definition of sexism, including discrimination based on sex, sexual orientation, and gender identity. We respectfully ask the reviewer to reconsider the evaluation in light of these contributions and our detailed responses to the concerns raised.
>
> **W1: Dataset size for training and future expansion.** We agree with the reviewer about the importance of larger-scale datasets for fine-tuning. In this paper, we focus on creating a dataset to evaluate multimodal models on sexism detection. With respect to the state of the art, ≈11 hours of content is consistent with prior high-quality datasets for sexism detection and related studies on hate speech detection. For example, the study “Sexism Identification on TikTok: A Multimodal AI Approach with Text, Audio, and Video” (Arcos et al., 2024) includes ≈13 hours of content, and “Detection of Hate Speech in Videos Using Machine Learning” (Wu et al., 2020) includes 300 videos. Additionally, MuSeD is carefully annotated, achieving *almost perfect* inter-annotator agreement (0.81–1.00).
>
> **W2: Limitations in audio feature utilization.** In our evaluation, we assess Gemini 2 Flash (Video) using raw video inputs, which include audio features such as tone and background sounds. The model achieves an accuracy of 68.92% when evaluated against the multimodal label (incorporating both text and visual information). Future work could explore the explicit integration of non-textual audio features to better isolate their contribution, which is feasible thanks to our three-level annotation framework—at the text, audio, and full video levels—designed for detailed multimodal analysis.
>
> **W3: Phrasing clarity.** Thank you for this observation. In the introduction, we clarify that we introduced an annotation framework, with separate annotations on text, audio, and video, to analyze the contribution of different modalities to the classification of content as sexist or non-sexist. We will refine the phrasing in the abstract and contributions section for greater clarity in the revised version.
>
> **W4: Small percentage of BitChute videos.** We agree that the percentage of videos from BitChute is relatively small. However, the goal of our experiment was to evaluate the reliability of the human annotations using the BitChute sample (lines 229–232) and compare the human annotations with the model’s classification (lines 314–316). Additionally, incorporating more videos from BitChute could produce an imbalance in the dataset distribution of sexist and non-sexist videos, given that BitChute is a low-moderation platform that mostly includes videos containing hate speech. Considering that sampling from BitChute should be done carefully, we plan to increase the BitChute percentage in the future.

---

> > ### Comment · Reviewer_xWDg · 2025-06-09
> >
> > After reading the rebuttal, the authors have addressed all my confusion. Therefore, I would like to raise my score from to 6,

---

> > > ### Author Response · Authors · 2025-06-10
> > >
> > > Thank you. We're glad to hear that you appreciated our clarifications.

---

### Decision · Program_Chairs · 2025-07-08

**Decision:**

Accept

**Comment:**

This paper presents the first multimodal Spanish video dataset for sexism detection (with a broad definition of sexism), which all reviewers acknowledge as a valuable contribution. The annotation framework was very well thought with high quality standards and agreement. Having annotations on text, audio, and video is definitely useful for exploring contributions of different modalities for multimodal models (although audio is relatively unexplored here). That said, the dataset is small (11 hours) for any sort of training / fine-tuning, and there are questions about the BitChute source (e.g. >90% of its videos are labeled sexist, although it only comprises 33 videos). The authors did respond to criticism about the prompts used in their experiments, which satisfied the reviewers. Overall, a strong dataset/benchmark paper that touches on some angles less commonly explored in mainstream LLM work.

**This paper went through ethics reviewing. Please review the ethics decision and details below.**
Decision: All good, nothing to do  or only minor recommendations